# The Aberrant Expression of Biomarkers and Risk Prediction for Neoplastic Changes in Barrett’s Esophagus–Dysplasia

**DOI:** 10.3390/cancers16132386

**Published:** 2024-06-28

**Authors:** Young Choi, Andrew Bedford, Simcha Pollack

**Affiliations:** 1Department of Pathology, Yale School of Medicine, 434 Pine Grove Lane, Hartsdale, NY 10530, USA; 2Department of Internal Medicine, Yale School of Medicine, Bridgeport Hospital, 267 Grant St., Bridgeport, CT 06610, USA; andrew.bedford@bpthosp.org; 3Department of Business Analytics Statistics, St. John’s University Tobin College of Business, Queens, NY 11423, USA; pollacks@stjohns.edu

**Keywords:** Barrett’s esophagus, BE–dysplasia, esophageal adenocarcinoma, biomarkers, diagnostic adjunct, risk prediction

## Abstract

**Simple Summary:**

The accurate diagnosis of Barrett’s esophagus (BE) and accurate grading of BE-associated dysplasia are essential for the optimal management of BE patients during surveillance. However, the diagnosis of BE may be missed, and the grading of BE–dysplasia has poor reproducibility and intra- and interobserver variability. Thus, well-defined biomarker testing remains indispensable. We conducted immunohistochemistry tests of a panel of biomarkers and discovered that MUC2 and TFF3 are specific diagnostic markers for BE. Moreover, the aberrant expressions of p53, p16, cyclin D1, Ki-67, beta-catenin, and MCM2 are significantly associated with LGD, HGD, and EAC histology with a high sensitivity and specificity and can be used to predict the risk of neoplastic changes and progression.

**Abstract:**

**Background**: Barrett’s esophagus (BE) is a pre-neoplastic condition associated with an increased risk of esophageal adenocarcinoma (EAC). The accurate diagnosis of BE and grading of dysplasia can help to optimize the management of patients with BE. However, BE may be missed and the accurate grading of dysplasia based on a routine histology has a considerable intra- and interobserver variability. Thus, well-defined biomarker testing remains indispensable. The aim of our study was to identify routinely applicable and relatively specific biomarkers for an accurate diagnosis of BE, as well as determining biomarkers to predict the risk of progression in BE–dysplasia. **Methods**: Retrospectively, we performed immunohistochemistry to test mucin 2(MUC2), trefoil factor 3 (TFF3), p53, p16, cyclin D1, Ki-67, beta-catenin, and minichromosome maintenance (MCM2) in biopsies. Prospectively, to identify chromosomal alterations, we conducted fluorescent in situ hybridization testing on fresh brush samples collected at the time of endoscopy surveillance. **Results**: We discovered that MUC2 and TFF3 are specific markers for the diagnosis of BE. Aberrant expression, including the loss and strong overexpression of p53, Ki-67, p16, beta-catenin, cyclin D1, and MCM2, was significantly associated with low-grade dysplasia (LGD), high-grade dysplasia (HGD), and EAC histology, with a relatively high risk of neoplastic changes. Furthermore, the aberrant expressions of p53 and p16 in BE-indefinite dysplasia (IND) progressor cohorts predicted the risk of progression. **Conclusions**: Assessing the biomarkers would be a suitable adjunct to accurate BE histology diagnoses and improve the accuracy of BE–dysplasia grading, thus reducing interobserver variability, particularly of LGD and risk prediction.

## 1. Introduction

Barrett’s esophagus (BE) is a pre-neoplastic condition associated with an increased risk of esophageal adenocarcinoma (EAC) [1,2,3]. The increased availability of specific biomarkers in BE and BE-associated dysplasia can optimize the management of patients with BE during surveillance, allowing for more aggressive endoscopic surveillance and treatment options for high-risk individuals, while avoiding frequent endoscopy in low-risk individuals [4,5,6,7]. Thus, well-defined biomarker testing remains indispensable for the accurate diagnosis of BE, as well as for predicting the risk of neoplastic progression and the therapeutic outcomes and survival of EAC patients [8,9,10,11,12,13].

The current diagnosis of BE in the USA requires histologic confirmation of intestinal metaplasia (IM) with the presence of true goblet cells (GCs) and BE for which true GCs were identified is essential in identifying BE-dysplasia [14]. However, the identification of true GCs on H&E sections is sometimes challenging and the level of inter-observer agreement in diagnosing true GC from pseudo-GC and IM of the gastric cardia is, on occasion, poor in the current practice setting. 

To enhance the accurate detection of GC and to establish the diagnosis of BE, various markers have been investigated using histochemistry and immunohistochemical stains [15]. The specificity of Alcian blue histochemical stain for GC was generally low with 72% of the positive predictive value. The immunohistochemistry stains of CDX2 were positive in 77% of IM and MUC2, in 78% with 98% specificity in non-GC columnar epithelium, indicating that both immunomarkers were superior to Alcian blue. Overall, Alcian blue, CDX2, and MUC2 stains were considered relevant in establishing the diagnosis of BE [15]. However, they were not entirely specific, and have not been adopted as a routine staining procedure in the clinical laboratory. Careful morphologic evaluation was considered the most useful way to determine the origin of IM near the gastro-esophageal junction.

Recently, studies have shown that concurrent BE was identified in 56.6% of patients with EAC [16]; Sawas et al. [17] found about 50% of EAC patients with a lack of IM or heartburn symptoms. In another retrospective cohort study, two-thirds of EACs are diagnosed at the time of BE diagnosis, suggesting the need for novel BE diagnostic strategies to reduce the mortality of EAC [18]. Undiagnosed BE in patients with EAC may be due to no symptoms or atypical symptoms during the initial disease progression or because the individuals with known risk factors are rarely referred for endoscopy. Overall, this represents a considerable missed opportunity for endoscopic surveillance. Further, Singer et al. [19] reported that screening of BE with endoscopy and forceps biopsies missed 50% of BE. The missed BE in the biopsy samples might be due to sampling errors and other diagnostic pitfalls including inflammatory changes, inflamed cardia due to reflux, gastric heterotopia, and metaplasia in the distal part of the esophagus. Further, the missed BE in the biopsies may also be related to the missed identification of GC on routine histology stains. Such missed BE would result in a missed opportunity for surveillance for early detection of neoplastic changes in BE and potential progression to EAC.

The importance of identifying BE was well demonstrated by Dhaliwal et al. [20] who found a lower rate of missed dysplasia in patients with the diagnosis of BE during endoscopic surveillance. Patients with EAC who had a prior diagnosis of BE had a survival advantage and better prognosis compared with those without a prior diagnosis of BE [21]. Thus, more accurate diagnostic biomarkers and strategies for BE can assist pathologists and clinicians in the early detection and optimum management of patients with BE with neoplastic progression of BE during surveillance, and reduce the morbidity and mortality of EAC.

MUC2, coding for the protein core (>5100 aa) of a major mucin secretion, is strongly expressed in intestinal goblet cells and is the diagnostic marker for IM of BE [22,23,24]. MUC2 staining in non-goblet columnar cells even without GCs is a sensitive and highly specific marker for intestinal-type mucin production and its presence is predictive of columnar metaplasia of the esophagus. In addition, trefoil factor (TFF) 3 contains a characteristic trefoil motif and consists of a 40 amino acid domain with three conserved disulfides. TFF3 is known to be specific for GCs [25,26,27,28].

The current management guidelines for the frequency of the endoscopic surveillance of BE are based on the grade of dysplasia [4,5]. Low-grade dysplasia (LGD) has a markedly increased risk of malignant progression to high-grade dysplasia (HGD) and EAC [2,3,29]. However, there is considerable intra- and interobserver variability and poor reproducibility in the diagnosis and grading of dysplasia [3,7,14,29,30,31,32,33], where LGD is overdiagnosed or underestimated [33,34]. A major diagnostic error of HGD was due to over-interpretation [35,36]. Thus, at least two pathologists should independently perform histological examinations in each case to avoid subjectivity in diagnosing dysplasia [37,38,39]. The number of pathologists who confirmed dysplasia by applying strict criteria was associated with the incidence and rate of progression [2,34,40,41]. Thus, the utility of these biomarkers in the process of molecular alterations predicting the earliest neoplastic changes, as well as the progression of the neoplastic transformation, may be useful adjuncts to histologic evaluation and improve the interobserver variability of histologic diagnoses [7,8,10,11,12].

The molecular progression of GERD-BE-EAC is a complex process involving multiple genetic alterations, including chromosomal aberrations, allelic imbalance, microsatellite instability, methylated DNA, and microRNA alteration causing consecutive deregulations of their products, cell cycle regulatory factors, the loss of heterozygosity (LOH) in locus 9p21 (involving gene CDKN2A) and locus 17p13 (TP53), the increase in c-Myc, the up-regulation of minichromosome maintenance 2 (MCM2) and cyclin D1, and reduced cell adhesion molecules [5,8,9,11,42,43,44]. Many potential immunohistochemical and molecular biomarkers have been investigated for the diagnosis of BE and objective methods in assessing the risk of progression of BE to BE-associated dysplasia. Studies have shown that a single or combination of biomarkers described below are considered to be involved in the molecular processes of GERD-BE-EAC at different stages of neoplastic changes. The markers may become the potential candidates to identify the neoplastic progression of BE in routine clinical laboratories.

### 1.1. Cell Cycle Regulatory Genes

The p53 tumor suppressor gene is shown to be one of the most frequent genetic alterations in BE-associated dysplasia. TP53 mutations often occur early in the non-dysplastic intestinal epithelium and amplification typically occurs as a late event [10]. Mutations of TP53 were identified in 72–82.6% of EAC [45,46] and were strongly associated with neoplastic progression, regardless of histologic diagnosis. p53 mutations are much more frequent in progressive cohorts than in non-progressive cohorts [10,47,48,49]. The overexpression and/or loss of p53 (aberrant expression) may arise long before the morphological detection of dysplasia in progressors [10]. Thus, aberrant p53 expression was identified as a risk factor for predictive and prognostic biomarkers in BE and BE-associated dysplasia [40,41,50]. Despite this, p53 testing for BE is not standard practice in the USA.

P16 is a member of the INK4 family of cyclin-dependent kinase inhibitors. The inactivation of p16 is an early and initiating genetic alteration, predicting the progression of BE and BE-associated dysplasia. It commonly occurs through the LOH of 9p21 or p16 methylation and, less commonly, via mutations [5,8,11,42,51]. The lack of p16 transportation to the nucleus and cytoplasmic accumulation of the protein via the inactivation of the p16 (INK4A/CDKN2A) tumor suppressor gene resulted in an aberrant expression of p16.

Cyclin D1 is a proto-oncogene that complexes with and activates cyclin-dependent kinases (CDK4/6). In a phosphorylated state, the retinoblastoma gene, which is highly expressed in BE-associated dysplasia, becomes inactive, allowing progression from the G1 to the S phase of the cell cycle and stimulating proliferation [8,52,53,54].

### 1.2. Proliferation Markers

MCM2 is a well-recognized group of proteins that can cause tumor initiation, proliferation, and progression by modulating the cell cycle and DNA replication stress. MCM2 is overexpressed at the luminal surface in BE, and its expression correlates with the degree of dysplasia and the tumor grades [8,9,55,56,57,58]. Ki-67 overexpression in BE is associated with the progression of BE to HGD and EAC. It is useful in distinguishing HGD and EAC from BE without dysplasia [11,57,59,60].

### 1.3. Cell Adhesion Markers

Beta-catenin mediates cell–cell adhesion via the transmembrane E-cadherin–catenin complex. In transformed cells, an excess of free (monomeric) beta-catenin binds to other newly synthesized proteins and is transferred to the nucleus, resulting in the down-regulation of E-cadherin and positive regulation of the epithelial–mesenchymal transition upon invasion [61]. Decreased beta-catenin expression was observed in gastrointestinal cancers [58,61,62,63].

### 1.4. Genomic Instability

For chromosomal alterations, fluorescent in situ hybridization (FISH)—utilizing four fluorescently labeled locus-specific probe sets targeting 8q24 (MYC), 9p21 (CDKN2A; alias p16), 17g12 (ERBB2; alias Her-2/neu), and 20q13 (ZNF217)—was able to differentiate HGD and EAC from other BE-associated dysplasia with a sensitivity of 80% and a specificity of 88% [64,65,66,67,68]. FISH can be performed on either formalin-fixed paraffin-embedded sections (FFPES) or fresh brushing specimens. The presence of polysomy may be able to serve as a confirmatory marker of HGD and EAC and as a screening tool to identify patients at the highest risk for the subsequent detection of HGD and EAC [69].

Given the limitation of the diagnosis of BE–dysplasia in biopsy samples, more sensitive and specific risk factors of progression of BE to BE–dysplasia are required for the diagnosis of BE and assess the risk of BE–dysplasia. Despite recent advances in the understanding of molecular biology and pathology of BE-associated neoplastic lesions and intensive evaluation of various markers [29,30,31,32,33,34,35,36,37,38,39,40,41,42,43,44,45,46,47,48,49,51,52,54,61,62,64,70], the markers studied and tested did not succeed in the direct transfer of the investigations to clinical practice; this may be related to technical challenges, requirements of special facility and resources, cost, and complex validation processes of molecular testing. At present, there is no biomarker testing adopted in a routine clinical laboratory.

Thus, in this study, we aimed to identify routinely applicable and cost-effective relatively specific biomarkers for the accurate diagnosis of BE, as well as predict the progression of BE and BE–dysplasia that stratify the management of patients during routine screening and surveillance at community endoscopy centers. Since IHC and simple molecular testing are routinely performed in most clinical laboratories, we reviewed the biomarkers that have been studied by others using IHC and FISH. We included most of the pertinent biomarkers involved in the neoplastic progression of BE (previously tested by other researchers), as well as the diagnostic biomarkers for BE. Given the complexity involved in the transformation of BE to EAC, a single biomarker may be inadequate for the prediction of progression in BE and BE-associated dysplasia. Retrospectively, we performed immunohistochemistry (IHC) for a panel of relatively specific biomarkers to determine their utility and clinical applicability to endoscopic biopsies: MUC2 and TFF3 for the diagnosis of BE, as well as p53, p16, cyclin D1, Ki-67, beta-catenin, and MCM2, to estimate the progression and prediction of neoplastic changes in BE. Prospectively, we tested FISH on fresh brush samples collected at the time of endoscopy surveillance to identify genetic alterations in patients with BE and BE-associated dysplasia.

## 2. Materials and Methods

The Institutional Review Board of Bridgeport Hospital (Bridgeport, CT, USA) approved the study protocol (#111304). The study cohort included 140 patients who were under surveillance care in a community cohort setting from 1999 to 2018. A total of 377 esophageal biopsies and 6 mucosal resections were included. The age of the patients ranged from 27 to 93 years, and the male-to-female ratio was 3:0 to 2:0. We examined and reviewed medical records of the patients to determine the length of follow-up, endoscopic findings, and clinical management. All had been diagnosed with GERD by clinical criteria and underwent serial endoscopy and biopsies.

### 2.1. Study Design

In a retrospective design, our study included patients with a confirmed diagnosis of BE at the baseline endoscopy biopsy and subsequent endoscopy biopsies with a follow-up of a minimum two-year surveillance period before the endpoint of the study. Eighty-five patients had multiple endoscopy procedures ranging from 2 to 10 follow-ups, with biopsies at different surveillance time points after the baseline endoscopy. Thus, the subsequent endoscopies could have had a different histologic diagnosis compared with the baseline biopsy diagnosis over time. We combined a series of all events from all patients and analyzed them together at the defined time points. The biopsy specimens were processed for a formalin-fixed paraffin-embedded section (FFPES) and then stained for H&E and immunohistochemistry (IHC). The histologic diagnoses (expert diagnoses) of all biopsies obtained at different time points were rendered by experienced pathologists with a subspecialty in gastrointestinal pathology. Each cohort was categorized as BE, BE-indefinite dysplasia (IND), low-grade dysplasia (LGD), high-grade dysplasia (HGD), and EAC based on the baseline histology diagnosis: 184 BE, 105 BE-IND, 61 LGD, 12 HGD, and 15 EAC. The LGD progressor (15) was defined as a baseline diagnosis of LGD followed by a diagnosis of HGD or EAC during follow-up. Furthermore, we evaluated the patients with BE-IND for the progression to LGD; 51 patients with BE-IND who progressed to LGD were defined as the BE-IND progressor cohort, and 47 patients who did not progress to LGD as the BE-IND non-progressor cohort. There were significant differences in baseline characteristics between the cohorts at large.

In a prospective design, at the time of endoscopy and biopsy, cytology specimens for FISH testing were collected in 48 patients with BE and BE–dysplasia (31 BE, 14 BE-IND, 3 LGD) by brushing the entire Barrett’s segment with a standard cytology brush (Cook Endoscopy, Winston-Salem, NC, USA) and by collecting cells through washing the brush with PreservCyt (Hologic) fixative solution.

### 2.2. Immunohistochemistry Study of Biomarkers

IHC was conducted for the protein expressions of MUC-2 (SPM513, Abcam, Cambridge, UK), TFF3 (EPR 3973, Abcam, Cambridge, UK), p53 (DO7, Dako, Carpentena, CA, USA), p16 (E6H4, Roche diagnostics, Indianapolis, IN, USA), cyclin Dl (EP12, Dako, Carpentena, CA, USA), B-catenin (CAT-15, ThermoFisher Scientific, Waltham, MA, USA), Ki-67 (MIB-1, Dako, Carpentena, CA, USA), and MCM2 (rabbit polyclonal, Abcam, Cambridge, UK) using the standard IHC protocol established in our laboratory with Bond-Max automatic immunostainer (Leica Biosystems, Deer park, IL, USA). Positive and negative tissue and reagent controls were included in each run. Positive staining was validated by positive tissue controls and a set of HGD patients. Each biopsy collected during the separate endoscopy procedure was individually evaluated and interpreted without the knowledge of the diagnosis of the biopsies taken at the prior or subsequent endoscopy. The interpretation of IHC scoring was performed blinded to morphologic diagnosis and outcome and progression status. The number of IHC-stained slides on each marker varied as the IHC-stained slide may not contain the representative lesion. The IHC was conducted by one histotechnologist who has extensive experience in IHC techniques with high reproducibility of IHC staining quality.

MUC2 and TFF3 immunoreaction was evaluated for the presence of true GCs by observing cytoplasmic perinuclear stains of MUC2 and the distinct strong oval-shaped cytoplasmic staining of TFF3 [23,25,26,28].

For p53 [49,59], cyclin D1 [56], Ki-67 [54,59], and MCM2 [49,59], the nuclear reaction was evaluated by combining the percentage and the intensity of nuclear staining. The strong and diffuse nuclear reaction present in the entire crypt(s), contiguous to the surface epithelial cells, was a 4+ overexpression, as well as a strong nuclear positivity present in >50% of cells in a crypt and/or one or more crypts, mostly at the basal crypts as 3+. The loss of p53, cyclin DI, MCM2, and Ki-67 was considered when dysplastic epithelial cells showed no nuclear reaction in one or more crypts. Heterogenous or weak staining in a few cells was interpreted as negative. The IHC reactions previously tested as a part of routine histology processing were re-reviewed by the author and all newly stained IHCs were evaluated and interpreted by the author without knowledge of clinical information or histology or previous IHC staining.

p16 immunostaining in both nuclear and block cytoplasmic reactions expressed in a continuous segment of crypts involving basal and parabasal layers extending to the surface epithelial cells (at least 10–20 cells) was interpreted as an aberrant reaction corresponding to 4+, and a separate cytoplasm or nuclear reaction in a few or several crypts corresponding to 3+ was also evaluated [70]. No reaction occurring in dysplastic cells was considered to be a loss of reaction. The strong positive reaction and loss of reaction were analyzed together in the statistical analyses. No immunoreaction of the cytoplasm or nuclei was considered to be negative. Beta-catenin staining was evaluated for the loss of membrane staining with a positive nuclear reaction or disrupted or reduced membrane stain or only nuclear staining. The presence of complete membrane staining was considered negative.

### 2.3. Fluorescent In Situ Hybridization (FISH)

The brush specimens for the FISH were sent to AcuPath laboratory (Plainfield, NY, USA) where the FISH test was conducted using Vysis Esophageal FISH Probe Kit, which consisted of four locus-specific identifier (LSI) probes: Spectrum Green ERBB2 (17q12), SpectrumRed p16 (9p21), SpectrumAqua-MYC (8q24), and SpectrumGold-ZNF2l7 (20q13.2) chromosomal loci. FISH tests were used to assess deletions, gains, or amplification of specific genes by evaluating 100 consecutive epithelia cells. FISH signal enumeration was performed without knowledge of the patient’s clinical or histological diagnosis and was interpreted as abnormal when one or more of the following criteria were met: >6% of cells had homozygous 9p21 loss, >11% of cells had homozygous or heterozygous 9p21 loss (combined), >4% of cells had polysomy, >5% of cells had tetrasomy, and >5% of cells gained a single locus.

### 2.4. Statistical Methods

Continuous variables were described with mean and standard deviations. Bivariate categorical testing compared histopathology and the degree of marker expression. Similarly, progressors/non-progressor status was predicted using the bivariate categorical association with the degree of marker expression. In all comparisons, the reference category of ‘negative’ expression was used. of ‘negative’ expression was used. Relative risk (RR) and its 95% confidence interval was computed to estimate and test relationships between categorical variables. The rates of progression to neoplasia (LGD, HGD, or EAC) and advanced neoplasia (HGD or EAC) were determined in the incident neoplasia group and reported as relative risk and odds ratio, as appropriate. The sensitivity and specificity for these 2 × 2 arrays were also reported.

In lieu of *p*-values, statistical significance was declared when the 95% confidence interval for the relative risk excludes 1. Wide confidence intervals, though significant, may indicate an insufficient number of events of interest and should be interpreted with caution. No adjustment for multiple analyses was made. Statistical calculations were performed using SAS 9.4 (SAS Institute Inc., Cary, NC, USA).

## 3. Results

### 3.1. Accurate Diagnosis of Barret’s Esophagus

MUC2 IHC exhibited distinct perinuclear and cytoplasmic staining of GCs in BE (Figure 1B). TFF3 IHC demonstrated strong cytoplasmic staining of small, large, and fully formed oval-shaped GCs (Figure 1C). Both TFF3 and MUC2 stains confirmed the histologic diagnosis of BE and were negative in the pseudo-GCs seen on the H&E stains, differentiating pseudo-GCs from true GCs. MUC2 and TTF3 both demonstrated the diagnostic accuracy of GCs identified using PAS or Alcian blue staining in 100% of stains. Both TFF3 and MUC2 was also used to identify additional GCs that were not identified using H&E staining or PAS or Alcian Blue staining in 15% of cases. MUC2 was reduced in 8.2% (5/61) of LGD (Figure 1D–F) or lost in 11.1% (4/33) of EAC (Figure 1G–I), while TFF3 was positive in all GCs of BE and BE-associated dysplasia. This indicates a high sensitivity and specificity of TFF3 in detecting GCs over MUC2.

### 3.2. The Rate of Aberrant Expression of Biomarkers in Barret’s Esophagus and BE-Associated Dysplasia

#### 3.2.1. p16 Expression

p16 expression in both nuclear and cytoplasmic expression (4+) and nuclear or cytoplasmic reactions (3+) was significantly associated with the degree of dysplasia (Table 1 and Figure 2): 10.7% (9/84) in BE, 23.1% (15/65) in BE-IND, 12.5% (1/8) in GERD, 64.4% (38/59) in LGD, 71.4% (15/21) in HGD, and 83.3% (10/12) in EAC. Strong nuclear and block cytoplasmic expressions in a continuous segments of crypt(s) from basal and parabasal layers extending to the surface epithelial cells were mostly present in HGD and EAC (Figure 2O). A loss of p16 staining was observed as follows: 11.9% (7/59) in LGD, 14.3% (3/21) in HGD, 8.3% (1/12) in EAC (Figure 3B,F), 1.5% (1/65) in BE-IND, and 0% in BE. Complete negative staining or blush weak cytoplasmic staining was observed as follows: 63.5% (55/84) in BE, 60.0% (39/65) in BE-IND, 8.5% (8/59) in LGD, 4.8% (1/21), and 0% in EAC.

#### 3.2.2. p53 Expression

Strongly positive nuclear staining (4+) involving the entire crypt(s) extending to the surface epithelium (Figure 2H,K,N) was identified as 63.5% (33/52) in LGD, 75% (9/12) in HGD, and 85.7% (6/7) in EAC, but only 17.2% (14/79) in BE-IND biopsies and 8.7% (5/90) in BE (Table 2). The predominant crypt pattern (3+) was confined either to the base or proliferative zones of the crypts and arranged in multifocal aggregates or in a scattered pattern (Figure 2H). It was present in 36.7% (33/90) of BE, 44.3% (35/79) of BE-IND, 13.5% (7/52) of LGD, 8.1% (1/12) of HGD, and 14.3% (1/7) of EAC. LGD revealed persistent p53 positive expression throughout the course of their follow-up period. p53 expression (Figure 3G,I) was lost as follows: 23.1% (12/52) in LGD, 25% (3/12) in HGD, 14.3% (1/7) in EAC, and 2.5% (2/79) in BE-IND.

#### 3.2.3. Alternate Loss or Overexpression of p53 or p16

While p16 is lost in BE-IND (Figure 3B) and HGD (Figure 3F), p53 is expressed, and while p53 is lost in LGD (Figure 3G) and EAC (Figure 3I), p16 is expressed.

#### 3.2.4. MCM2, Cyclin DI, and Ki-67 Expression

The overexpression of cyclin D1, MCM2, and Ki-67 showed a similar pattern of nuclear reactions and progressively increased with the severity of dysplasia (Table 2). The strong nuclear expression (at least >50% cells) of the crypt extending from the basal layer to the superficial epithelium was mostly observed in LGD, HGD, and EAC (Figure 2D–F).

A strong positive MCM2 nuclear reaction (4+) was detected as 80.0% (44/51) in LGD, 100% (11/11) in HGD, 90.1% (10/11) in EAC, 7.4% (4/54) in BE, and 15.9% (7/44) in BE-IND, indicating that the overexpression of MCM2 is a differentiating marker for LGD, HGD, and EAC from BE and BE-IND (Table 2). The predominant crypt pattern of MCM2 reaction (3+) was observed in 48.6% (25/54) of BE, 52.3% (23/44) of BE-ITD, 9.8% (5/51) of LGD, and 0% of HD and EAC. MCM2 reaction was lost in 3.9% (2/51) of LGD, 9.0% (1/11) of EAC, and 0% of HGD. While p53 was lost in dysplastic cells (Figure 3B), MCM2 was retained in 10% of HGD. MCM2 may be a useful marker for detecting BE–dysplasia.

The overexpression of cyclin D1 (4+) was observed as follows: 69.8% (30/43) in LGD, 81.8% (9/11) in HGD, 77.8% (7/9) in EAC (Table 2), and 0% in BE and BE-IND. The predominant crypt pattern (3+) of cyclin D1 was observed as follows: 26.5% (13/49) in BE, 56.4% (22/39) in BE-IND, 18.6 (8/43) in LGD, and 27.3 (3/11) in HGD. The cyclin D1 nuclear reaction was lost in 18.6% (8/43) of LGD and 18.2% (2/11) of HGD. A weakly positive cyclin D1 nuclear reaction (1–2+) in a few cells was observed as follows: 40.8% (20/49) in BE, 17.9% (7/39) in BE-ITD, 4.7 (2/43) in LGD, and 0% in HGD and EAC. A strong positive diffuse Ki-67 nuclear reaction (4+) extending to the surface columnar cells served as a consistent discriminator of LGD from HGD. Ki-67 in BE and BE-IND showed a few scattered nuclear Ki-67 reactions that were <10% of a crypt.

#### 3.2.5. Beta-Catenin Expression

Aberrant beta-catenin staining with the increased expression of nuclear and cytoplasmic reaction (4+) occurred exclusively in LGD, HGD, and EAC (Figure 4G): 42.2% (14/33) in LGD, 50.0% (10/20) in HGD, 66.6% (2/3) in EAC, 4.5% (1/22) in BE-IND, and 0% in BE (Table 1). The nuclear accumulation of beta-catenin was associated with the transition from BE to BE-associated dysplasia. Condensed and disrupted or reduced membrane staining was observed as follows: 55.6% (25/45) in BE, 68.2% (15/22) in BE-IND, 45.5% (15/33) in LGD, and 25.0% (5/20) in HGD. Loss of membrane stains was demonstrated in 9.1% (3/33) of LGD, 25.0% (5/20) of HGD, 33.3% (1/3) of EAC, 4.5% (1/22) of BE-IND, and 0% of BE. Complete membrane staining (negative reaction) was present in 44.5% (20/45) of BE, 9.1% (2/22) of BE-IND, 3.0% (1/33) of LGD, and 0% of HGD and EAC.

### 3.3. Fluorescent In Situ Hybridization (FISH)

FISH testing showed a variety of chromosomal alterations in 25.0% (12/48) of patients tested: p16 deletion in 33.3% (two BE and two BE-IND), >4% of tetrasomy in 33.3% (two LGD with >10% tetrasomy; one BE and one BE-IND with >5% tetrasomy), and polysomy in 33.3% (two BE and two BE-IND). More than one chromosomal abnormality was observed in 58.3% (7/12). A patient with BE-IND with three consecutive abnormal FISH tests is included in the Appendix A.

#### Concurrent Chromosomal Alterations by FISH and Aberrant p53 and p16 Protein Expression

Abnormal FISH testing in two patients with LGD showed the concurrent aberrant expression of p16 and p53 by IHC: one LGD patient with an overexpression of p53, loss of p16 expression and >10% tetrasomy, and another LGD patient with a loss of p 16 protein expression and >10% tetrasomy. In two patients with BE-IND, the deletion of p16 in FISH showed a loss of p16 protein expression and predominant p53 overexpression in the crypt.

### 3.4. Significant Association of Aberrant Expression of Biomarkers and Relative Risk of Neoplastic Changes in BE–Dysplasia

As shown in Table 3, aberrant expressions, including the strong overexpression and loss of expressions of p53, p16, MCM2, beta-catenin, and cyclin D1, were significantly associated with LGD histology, with a high sensitivity ranging from 94 to 98% and specificity ranging from 87 to 95%, as well as a high RR of neoplastic changes. Combined LGD, HGD, and EAC showed similar findings.

The aberrant p16 reaction of both nuclear and cytoplasmic reactions (4+) showed a significant association with LGD histology, with a 90% sensitivity, 87% specificity, and an RR of 10.8 (95% CI, 4.6–25.4). It was also significantly associated with BE-IND histology, with an 87% specificity, 29% sensitivity, and an RR of 1.6 (95% CI, 1.1–2.4). Aberrant p53 reaction showed a significant association with the diagnosis of LGD, with a 98% sensitivity, 91% specificity, and an RR of 47.7 (95% CI, 6.8–333.1). It was also significantly associated with the diagnosis of BE-IND in a 91% specificity but 33% sensitivity and with an RR of 2.0 (95% CI, 2–2.8). A strong p53 overexpression colocalized to BE-IND histology seems a positive predictor of the eventual progression to LGD.

The overexpression (4+) or aberrant reaction of cyclin D1 and MCM2 biomarkers also demonstrated a significant association with LGD histology: aberrant MCM2 expression had an RR of 24.8 (95% CI, 3.6–170.3), 83% sensitivity, and 87% specificity, while cyclin D1 had an RR of 1 6.5 (95% CI, 2.4–10.6), 97% sensitivity, and 94% specificity. The aberrant expression of beta-catenin was significantly associated with LGD and with an RR of 19.8 (95% CI, 2.9–134.7) and with 94% sensitivity and 95% specificity.

Furthermore, the overexpression of p53 at the basal crypts (3+) was significantly associated with LGD histology, with an RR of 9.3 (95% CI, 1.2–72.4), 88% sensitivity, and 61% specificity. p16 nuclear or cytoplasmic expression was associated with LGD histology, with an RR of 6.4 (95% CI, 2.5–15.8), 74% sensitivity, and 80% specificity. Overexpression at the basal crypts (3+) of cyclin DI was significantly associated with LGD with 89% sensitivity and 55% specificity. A disrupted/reduced beta-catenin reaction was significantly associated with LGD and with an RR of 7.8 (95% CI, 1.1–55.6), with 94% sensitivity and 44% specificity. MCM2 overexpression at the basal crypts was not associated with LGD.

### 3.5. Aberrant Expression of p53 and p16 in LGD and BE-IND Progressors and Non-Progressor Cohorts

As shown in Table 4, aberrant expression including strong overexpression and loss of expression are significant risk predictors of progression in the LGD (15) and BE-IND progressor cohorts (5I).

In LGD progressors to HGD, an aberrant p53 reaction occurred in 80% (12/15) of progressors, and in 39.5% (15/38) of non-progressors with an RR of 3.1 (95% CI 0.2–41.9) but was not significantly associated with progression. A p16 aberrant reaction was detected in 80% of progressors (12/15), and 72.2% (26/36) of non-progressors, with an RR of 3.5 (95% CI, 0.2–50.9), but was not associated with progression.

On the other hand, in BE-IND progressor cohorts to LGD, p53 aberrant reaction was detected in 26.9% (14/52) of progressors, and in 2.6% (2/77) of non-progressors, with an RR of 4.4 (95% CI, 2.7–7.1). A p16 aberrant reaction was detected in 33.3% (17/51) of progressors, and in 9.0% (8/88) of non-progressors, with an RR of 4.4 (95% CI, 2.5–7.6). The aberrant expressions of p53 and p16 reaction in BE-IND progressor cohorts were significantly associated with progression from BE-IND to LGD, suggesting the aberrant p53 and p16 expressions in BE-IND to be risk predictors of progression from BE-IND to LGD.

## 4. Discussion

For the optimal management of patients with BE, the accurate diagnosis and early detection of BE is of utmost importance. In our study, the IHC of TFF3 and MUC2 confirmed the diagnosis of BE by identifying true GCs [23,24,25,26,27,28]. While MUC2 expression is lost in 11.1% of HGD and EAC, TFF3 was significantly expressed in true GCs, differentiating pseudo-GCs from true GCs. TFF3 is the sensitive and specific marker for the intestinal metaplasia of the esophagus. Identifying true GCs can trigger index endoscopy [18], and can detect prevalent EAC at an early stage, providing a better survival advantage [19,20,21]. Overall, MUC2 and TFF3 are highly specific and a valuable adjunct to histology that can be routinely applicable in a clinical laboratory setting.

The frequency of the endoscopic surveillance of BE is based on the grade of dysplasia, particularly low-grade dysplasia (LGD), as LGD has a markedly increased risk of malignant progression to high-grade dysplasia (HGD) and EAC. Thus, biomarkers predicting early neoplastic changes to LGD are essential. Our study demonstrated that aberrant expression, including the loss and strong overexpression of p53, Ki-67, p16, beta-catenin, cyclin D, and MCM2 increases with the severity of neoplastic changes and is significantly associated with LGD, a high relative risk, and high sensitivity and specificity.

Aberrant p16 expression is significantly associated with LGD, HGD, and EAC histology with 10.8 RR, 90% sensitivity, and 87% specificity (Table 3). The loss of p16 expression was exclusively demonstrated in LGD, HGDs, and EAC, but no loss was observed in BE and BE-IND (Table 1). The findings are consistent with other studies [5,8,42,70,71,72,73]. In the progression risk analysis, aberrant p16 showed a 3.5 RR but not significantly associated with LGD progressor cohorts, while it is significantly associated with BE-IND progressors in a 4.4 ( 95% CI, 2.5-7.6).

The overexpression of p53 was identified in all categories of BE-associated dysplasia but the degree of aberrant expression increased with the progression of dysplasia (Table 2). As shown in Table 3, aberrant p53 expression is significantly associated with LGD, HGD, and EAC and with 98% sensitivity and 91% specificity and RR of 47.7 in LGD, alerting to a potential risk of progression. Furthermore, as shown in Table 4, aberrant p53 expression showed a 3.1 RR but not significantly associated with the progression of LGD to HGD, while it is significantly associated with BE-IND to LGD in a 4.4 RR ( 95% CI, 2.7–7.1). Overall, aberrant p53 expression seems to serve as a marker for the likelihood of progression to HGD/EAC, as explored in other studies [49,50,74]. The loss of p53 was observed as 23% in LGD, 25% in HGD, 14.3% in EAC, and 0% in BE or BE-IND in our study. Other studies [75,76] also reported that the loss of p53 is a predictor of EAC [41,48,60,76,77,78,79]. The evaluation of p53 expression by IHC demonstrated improved reproducibility in morphological assessments and aids in avoiding the over- or under-diagnosis of dysplasia [41,48]. p53-IHC may reduce the rate of BE-IND histology diagnosis and improve the interobserver variability among pathologists when reporting BE with equivocal epithelial changes or BE-IND [36,76,80]. p53 immunostaining may be useful as a discriminative test to improve risk stratification, and hence the cost-effectiveness of BE surveillance programs, and more accurate risk stratification for individuals with considerable risk can be treated, while low-risk patients could continue with endoscopic follow-up. The aberrant expression of p53 in dysplastic Barrett’s epithelium may be an indication for the endoscopic treatment of the diseased mucosa [81]. Strong positive nuclear immunostaining (4+) of Ki-67 in the crypts extending to the surface epithelium is mostly detected in HGD and EAC, distinguishing HGD from LGD [60,61,82,83,84,85,86,87]. However, the rate of 4+ Ki-67 reaction was seen only in 20% of the biopsies stained in our study. Strong 4+ MCM2 overexpression can differentiate from BE to LGD, HGD, and EAC (Table 3), as explored in other studies [8,9,28,53,54,55]. Aberrant MCM2 expression is significantly associated with LGD, HGD, and EAC, with 99% sensitivity, 87% specificity, and an RR of 24.8 (Table 3). The quantification of MCM2 in combination with Ki-67 at the mucosal surface of both dysplastic BE and EAC is useful for recognizing patients with BE that is prone to develop into EAC [57].

Strong cyclin D1 reaction (4+) increased with the severity of dysplasia (Table 4) and is significantly associated with LGD, HGD, and EAC diagnoses, with 97% sensitivity, 94% specificity, and an RR of 16.6, which is similar to other studies [8,53,56,88]. The diffuse cytoplasmic reaction and the enhanced nuclear expression and/or complete loss of membranous expression of beta-catenin were significantly associated with LGD, HGD, and EAC (Table 3), with 94% sensitivity, 95% specificity, and an RR of 19.8. Reduced beta-catenin staining was observed in the early stage of the neoplastic progression of BE [53,57,61,62,89].

In BE-IND, only aberrant p53, cyclin D1, and p16 reactions were significantly associated with the BE-IND histology but with a low specificity. Other studies also demonstrated that aberrant p53 and p16 expressions are associated with potential risk stratification in BE-IND and LGD [90] and might benefit from more intensive surveillance or endoscopic eradication therapy [50].

FISH testing demonstrated abnormal chromosomal changes such as an increase in c-Myc, polysomy, and tetrasomy, and a loss of p16 in a small number of cases. The effectiveness of FISH testing in BE-associated dysplasia requires further study with a large cohort. In addition, FISH testing requires highly skilled personnel to perform and interpret the findings.

Recently, TissueCypher was introduced for mutational load assessment (BarreGen) testing. TissueCypher uses a multiplexed fluorescence imaging platform that automatically extracts quantitative data on multiple tissue biomarkers (p16, AMACR, p53, CD68, COX-2, CD45RO, HIF-lα, HER2, and CK20) (K20) on FFPES, and a multivariable classifier integrates quantitative image analysis data to provide a risk score, which classifies patients into categories of high, intermediate, and low risk of progression over 5 years [91,92,93]. The test is highly specific but currently only available in commercial laboratories and is expensive. DNA content abnormalities identified using DNA flow cytometry can support dysplasia diagnosis and aid in risk assessments for the development of HGD/EAC. However, these tests are not routinely applicable in clinical laboratories.

To adopt biomarker testing for the management of BE and BE–dysplasia, the testing should be technically simple and cost-effective with a high specificity and sensitivity, and should be routinely applied to available tissue materials. As p53 is one of the more common biomarkers, validated and routinely used in many laboratories, it can be easily integrated into routine practices for BE and BE–dysplasia diagnoses.

In summary, the aberrant expression of the biomarkers in LGD appears to be a useful adjunct to the routine histology, thus reducing the interobserver variability of LGD diagnosis in routine histology. Even crypt-based reaction (3+) of p53, cyclin D1, and MCM2, cytoplasm or nuclear reaction of p16, or disrupted beta-catenin reaction were significantly associated with the LGD, HGD, and EAC histology. The findings in our study support those of previous studies [8,9,11,12,47].

The limitations of our study include a limited number of cohorts and the absence of both matched controls of ages, sex, length of BE, smoking, and other clinical data in each cohort and a multivariate analysis. Furthermore, as we collected cases from a highly specialized tertiary care center, referral bias may have been present.

## 5. Conclusions

Our study provides the use of both histology and biomarker testing to support histology diagnosis and identify a subset of patients with BE–dysplasia who may be at increased risk of neoplastic changes. MUC2 and/or TFF3 are easily applicable confirmatory markers for the diagnosis of BE, and can be used to avoid misdiagnosis.

The aberrant expression of one or a combination of useful biomarkers with high sensitivity and specificity, such as cyclin D1, MCM2, p16, beta-catenin, Ki-67, and p53, appears to be suitable for use in a routine test in a clinical laboratory. Among these biomarkers, p16, p53, and/or MCM2 markers were associated with expert LGD, HGD, and EAC, demonstrating the most consistent reactions and risk predictions. They can serve as suitable adjuncts to routine histology in the grading of dysplasia, particularly for the risk stratification of BE-IND and LGD.

The British Society of Gastroenterology recommends assessing TP53 using IHC to solve the issue of equivocal histologic diagnoses of dysplasia, reflexively adding this to routine practice [94]. The findings of our study support the use of p53 IHC as a routinely applicable biomarker in BE-IND and LGD to predict the risk of neoplastic changes during surveillance in the USA. Our study highlights the need for more precise approaches to screen and identify individuals at risk of BE for the improved surveillance and early stage of neoplastic progression to EAC by adopting routinely available marker testing for risk stratification to delineate low- and high-risk populations.

## Figures and Tables

**Figure 1 cancers-16-02386-f001:**
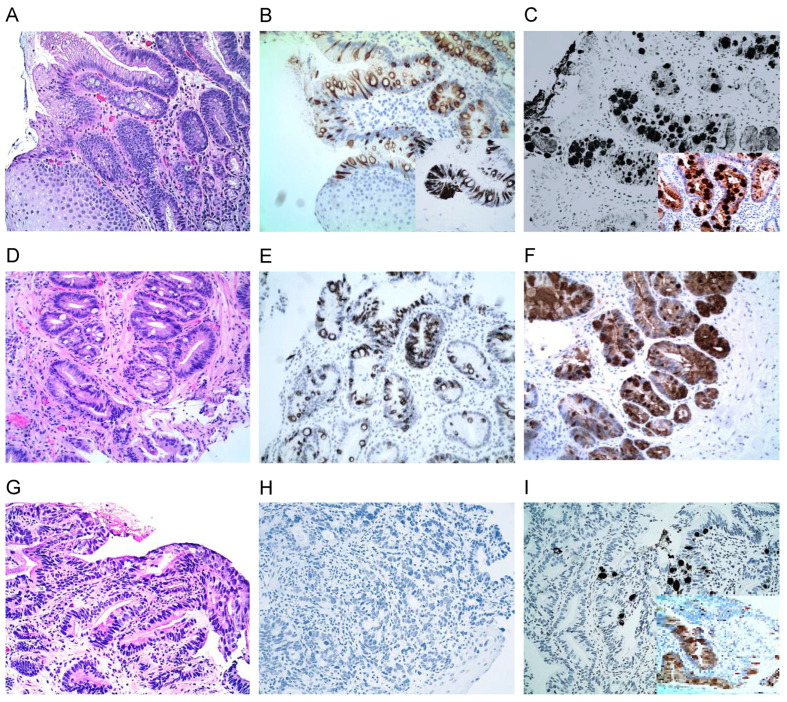
Immunochemistry stains of cytoplasmic or perinuclear reactions of MUC2 and the strong oval-shaped cytoplasmic staining of TFF3 in the goblet cells (GCs) in BE (**A**–**C**), LGD (**D**–**F**), and HGD (**G**–**I**). While MUC 2 in GCs is distinctly expressed in BE (**B**) and in LGD (**E**), it is lost in HGD (**H**). On the other hand, TFF3 in GCs is expressed in all BE (**C**), LGD (**F**), and HGD (**I**): (10× original magnification in (**A**–**I**); inserts in (**B**,**C**) and I with 20× original magnification).

**Figure 2 cancers-16-02386-f002:**
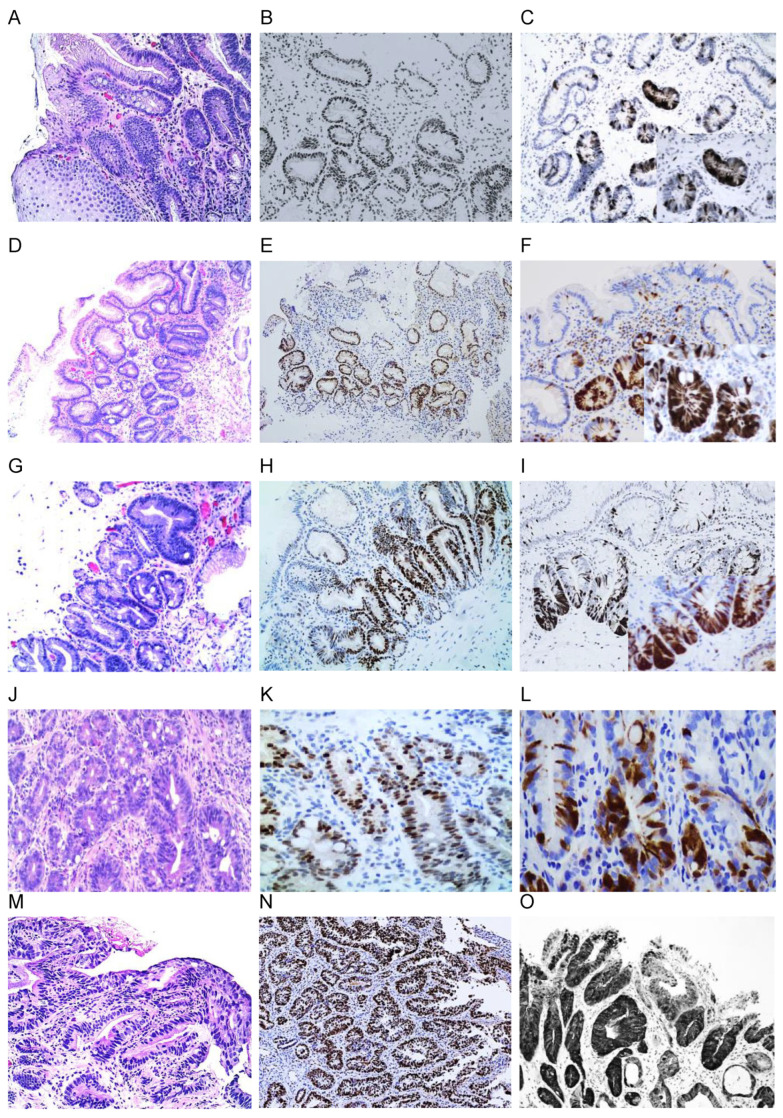
Immunohistochemistry stains of p16 and p53 in BE (**A**–**C**), BE-IND (**D**–**F**), crypt dysplasia (**G**–**I**), LGD (**J**–**L**), and HGD (**M**–**O**): p16 immunoreaction of both nuclear and cytoplasmic expressions in LGD (**L**) and HGD (**O**) and nuclear and/or cytoplasmic reaction in BE (**C**) and BE-IND (**F**); A strongly positive p53 nuclear reaction is associated with the degree of dysplasia: crypt predominant reaction in BE (**B**), BE-IND (**E**), crypt dysplasia (**H**), and a diffuse strong nuclear reaction in LGD (**K**) and in HGD (**K**): (10× original magnification; inserts in C and F with 20× original magnification).

**Figure 3 cancers-16-02386-f003:**
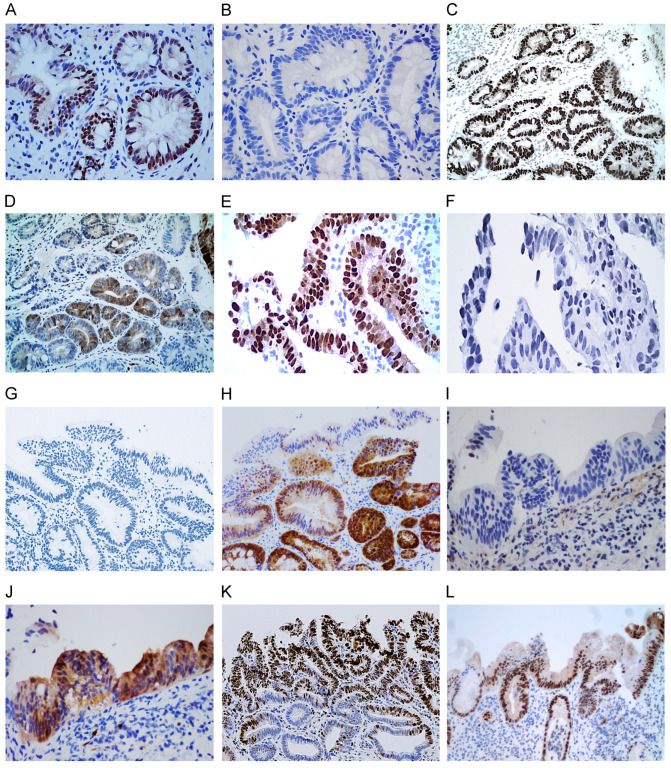
Differential expression of p16 or p53: expression of p53 (**A**) and loss of p16 (**B**) in BE-IND; expression of p53 (**C**) and reduction in p16 (**D**) in LGD; expression of p53 (**E**) and loss of p16 (**F**) in HGD; loss of p53 (**G**) and expression of p16 (**H**) in LGD; loss of p53 (**I**) and expression of p16 (**J**), MCM2 (**K**), and cyclin D1 (**L**) in EAC (20× original magnification).

**Figure 4 cancers-16-02386-f004:**
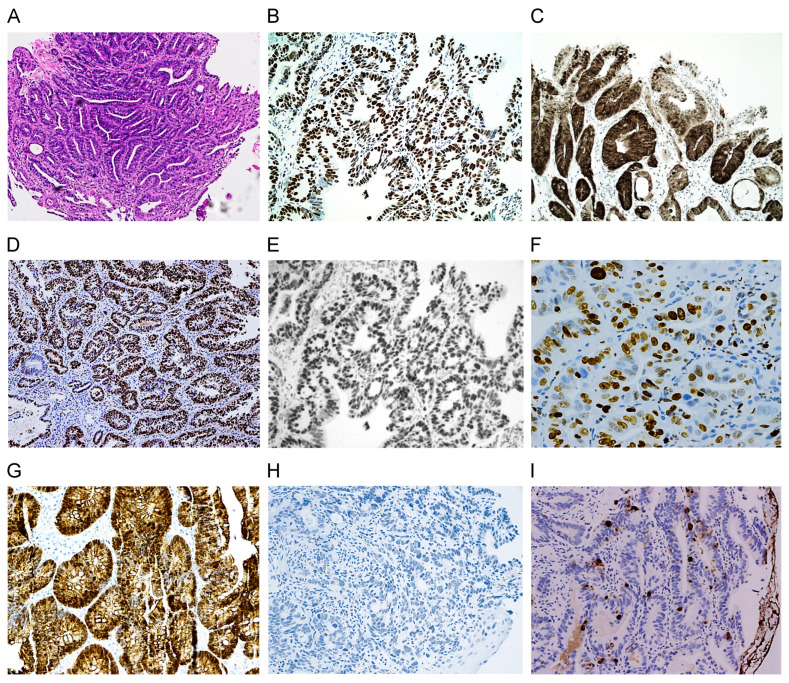
Immunohistochemistry stains of biomarkers in a case of EAC: H&E (**A**), p53 (**B**), p16 (**C**), MCM2 (**D**), Cyclin D1 (**E**), Ki-67 (**F**), Beta-catenin (**G**), MUC2 (**H**), and TFF3 (**I**). While MUC2 (**H**) is lost, TFF3 (**I**) is expressed (10× original magnification; inserts in B and J with 20× original magnification).

**Table 1 cancers-16-02386-t001:** Immunoreaction of p16 and beta-catenin in Barrett’s esophagus and BE–dysplasia.

Histopathology	Types and Degree of p16 Reaction				Types and Degree of Beta-Catenin Reaction			
	C and N *	C or N	Loss	Negative	C and N *	Disrupted/Reduced	Loss	Negative
	% (Case)	% (Case)	% (Case)	% (Case)	% (Case)	% (Case)	% (Case)	% (Case)
BE	10.7 (9/84)	23.8 (20/84)	0 (0/84)	65.5 (55/84)	0 (0/45)	55.6 (25/45)	0 (0/45)	44.5 (20/45)
BE-IND	23.1 (15/65)	15.4 (10/65)	1.5 (1/65)	60.0 (39/65)	4.5 (1/22)	68.2 (15/22)	4.5 (1/22)	9.1 (2/22)
LGD	64.4 (38/59)	23.7 (14/59)	11.9 (7/59)	8.5 (5/59)	42.2 (14/33)	45.5 (15/33)	9.1 (3/33)	3.0 (1/33)
HGD	71.4 (15/21)	9.5 (2/21)	14.3 (3/21)	4.8 (1/21)	50 (10)	25.0 (5/20)	25 (5/20)	0 (0/20)
EAC	83.3 (10/12)	8.3 (1/12)	8.3 (1/12)	0 (0/12)	66.6 (2/3)	0 (0/3)	33.3 (1/3)	0 (0/3)

* C and N: cytoplasm and nucleus; BE: Barrett’s esophagus; BE with IND; BE with indefinite dysplasia; LGD: low-grade dysplasia; HGD: high-grade dysplasia; EAC: esophageal adenocarcinoma.

**Table 2 cancers-16-02386-t002:** Immunoreaction of p53, MCM2, and Cyclin D1 in Barrett’s esophagus and BE–dysplasia.

Histopathology	Degree of p53 Immunoreaction				Degree of MCM2 Immunoreaction				Degree of Cyclin D1 Immunoreaction				
	4+ *	3+ **	Loss	Negative	4+ *	3+ **	Loss	Negative	4+ *	3+ **	1–2+ ***	Loss	Negative
	% (Case)	% (Case)	% (Case)	% (Case)	% (Case)	% (Case)	% (Case)	% (Case)	% (Case)	% (Case)	% (Case)	% (Case)	% (Case)
BE	8.7 (5/90)	36.7 (33/90)	0 (0/90)	57.8 (52/90)	7.4 (4/54)	48.6 (24/54)	0 (0/54)	48.1 (26/54)	0 (0/49)	26.5 (13/49)	40.8 (20/49)	0 (0/49)	32.7 (16/49)
BE-IND	17.2 (14/79)	44.3 (35/79)	2.5 (2/79)	41.7 (33/79)	15.9 (7/44)	52.3 (23/44)	0 (0/44)	31.8 (14/44)	0 (0/39)	56.4 (22/39)	17.9 (7/39)	0 (0/39)	25.6 (10/39)
LGD	63.5 (33/52)	13.5 (7/52)	23.1 (12/52)	1.9 (1/52)	80.0 (44/51)	9.8 (5/51)	3.9 (2/51)	0 (0/51)	69.8 (30/43)	18.6 (8/43)	4.7 (2/43)	6.9 (3/43)	0 (0/43)
HGD	75 (9/12)	8.3 (1/12)	25 (3/12)	16.7 (2/12)	100 (11/11)	0 (0/11)	0 (0/11)	0 (0/11)	81.8 (9/11)	18.2 (2/11)	0 (0/11)	0 (0/11)	0 (0/11)
EAC	85.7 (6/7)	14.3 (1/7)	14.3 (1/7)	14.3 (1/7)	90.1 (10/11))	0 (0/11)	9.0 (1/11)	0 (0/11)	77.8 (7/9)	0 (0/9)	0 (0/9)	22.2 (2/9)	0 (0/9)

* strong and diffuse reaction in the entire gland; ** strong but focal, mostly in the basal glands; *** few scattered positive nuclei; BE: Barrett’s esophagus; BE-IND: BE with indefinite dysplasia; LGD: low-grade dysplasia; HGD: high-grade dysplasia; MCM2: minichromosome maintenance2.

**Table 3 cancers-16-02386-t003:** Significance association for the aberrant expression of biomarkers and relative risk of neoplastic changes in Barrett’s esophagus–dysplasia.

Biomarkers	Histology Category	Degree of Expression	RR (95% CI)	Sensitivity	Specificity
**p53**	BE-IND	4+ *	2.0 (1.2–2.8) +	33	91
		3+ *	1.3 (0.93–1.9)	51	61
	LGD	4 *	47.7 (6.8–333.1) +	98	91
		4+ **	25.1 (3.6–173.9) +	98	67
		3+ *	9.3 (1.2–72.4) +	88	61
	LGD/HGD/EAC	4+ *	12.98 (5.0–33.5) +	94	91
		4+ **	7.4 (2.9–18.7) +	94	67
		3+ *	3.0 (1.0–9.1) +	69	61
**MCM2**	BE-IND	4+ *	1.8 (0.9–3.4)	33	87
		3+ *	1.4 (0.8–2.3)	62	52
	LGD	4+ *	24.8 (3.6–170.3) +	83	87
		4+ **	13.1 (1.9–86.7) +	98	67
		3+ *	4.6 (0.6–37.3)	95	59
	LGD/HGD/EAC	4+ *	25.5 (3.7–174.6) +	99	87
		4+ **	13.6 (2.0–90.5) +	99	67
		3+ *	4.6 (0.6–37.3)	83	52
**p16**	BE-IND	C&N and loss *	1.62 (1.1–2.4) +	29	87
		C or N	1.01 (0.6–1.7)	20	80
	LGD	C&N and loss *	10.8 (4.6–25.4) +	90	87
		C&N and loss **	6.5 (2.8–15.0) +	90	71
		C or N	6.3 (2.5–15.8) +	74	80
	LGD/HGD/EAC	C&N and loss *	9.8 (4.6–21.1) +	93	87
		C&N and loss **	6.2 (2.9–13.1) +	93	71
		C or N	5.84 (2.5–13.4) +	74	80
**Beta-catenin**	BE-IND	C&N and loss *	3.3 (1.1–10.2) +	29	95
		disrupted/reduced	1.87 (0.8–4.5)	75	44
	LGD	C&N and loss *	19.8 (2.9–134.7) +	94	95
		C&N and loss **	5.4 (0.9–32.2)	94	71
		disrupted/reduced	7.87 (1.1–55.6) +	94	44
	LGD/HGD/EAC	C&N and loss *	20.4 (3.0–130.4) +	97	95
		C&N and loss **	5.7 (0.9–34.0)	97	71
		disrupted/reduced	9.3 (1.3–64.9) +	95	44
**Cyclin D1**	BE-IND	4+ *	1.3 (0.3–5.6)	9	94
		3+ *	1.63 (0.9–2.8)	69	55
	LGD	4+ *	16.5 (2.4–110.6) +	97	94
		4+ **	16.7 (1.6–69.2) +	97	91
		3+ *	6.5 (0.9–46.8)	89	55
	LGD/HGD/EAC	4+ *	16.6 (2.5–111.7) +	98	94
		4+ **	10.8 (1.7–69.9) +	98	91
		3+ *	7.4 (1.0–52.4) +	91	55

* Reference is Barrett’s esophagus; ** reference is BE-IND; RR with + *p* < 0.05; C: cytoplasmic expression; N: nuclear expression; BE-IND: Barrett’s esophagus with indefinite dysplasia; LGD: low-grade dysplasia; EAC: esophageal adenocarcinoma; HGD: high-grade dysplasia.

**Table 4 cancers-16-02386-t004:** Aberrant expression of p53 and p16 in LGD progressors and Barrett’s esophagus–indefinite dysplasia progressor cohorts.

	Low-Grade Dysplasia to High-Grade Dysplasia				BE-IND * to Low-Grade Dysplasia			
Biomarkers	Progressors (%)	Non-Progressors (%)	RR (95% CI) **		Progressors (%)	Non-Progressors (%)	RR (95% CI) **	
**p53**								
4+ and loss	12 (80.0)	15 (39.5)	3.1 (0.2–41.9)		14 (26.9)	2 (2.6)	4.4 (2.7–7.1) +	
3+	3 (20.0)	20 (52.6)	1.01 (0.6–1.6)		23 (44.2)	15 (19.5)	3.0 (1.8–5.1) +	
Absent (ref)	0	3 (7.9)			15 (28.9)	60 (77.9)		
	15	38			52	77		
**p16**								
C/N and loss	12 (80)	26 (72.2)	3.5 (0.2–50.9)		17 (33.3)	8 (9.0)	4.4 (2.5–7.6) +	
C or N	3 (20)	5 (13.9)	4.1 (0.3–67.2)		20 (39.2)	10 (11.4)	4.3 (2.5–7.4) +	
Absent (ref)	0	5 (13.9)			14 (27.5)	76 (79.6)		
	15	36			51	88		

* BE-IND; Barret’s esophagus with indefinite dysplasia, ** RR with +; *p* < 0.05, C, N: cytoplasmic and nuclear expression.

## Data Availability

The manuscript includes datasets generated during and/or analyzed during the current study. Any additional data are available from the corresponding author upon reasonable request.

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
