# Peer review of "The Aberrant Expression of Biomarkers and Risk Prediction for Neoplastic Changes in Barrett’s Esophagus–Dysplasia"

_cancers, 2024, doi:10.3390/cancers16132386_

Round 1
Reviewer 1 Report
Comments and Suggestions for Authors
In the present study, Dr. Choi and colleagues conduct an in-depth study looking at IHC and FISH to diagnose Barrett’s esophagus, provide accurate grading, and prognosticate progression to higher grades. This is an important area of study and the authors are applauded on their efforts in this space. I have several suggestions/comments to improve the manuscript:
Abstract:
1. Please reorganize the abstract into a structured format, to be consistent with author guidelines for this journal (Background, Methods, Results, Conclusion).
2. Please ensure that all acronyms in the abstract are defined within the abstract itself
Introduction:
1. The premise statement that 20% of HGD/EAC diagnoses are missed in ~20% of the antecedent surveillance biopsy within a year is a difficult assertion to make based on available data, and is also different than the clinical question at hand (e.g., what can be done lower the rate of missed BE, and HGD/EAC diagnoses on surveillance endoscopy, which is at most 5% in large population-based analyses such as the one by Dhaliwal and colleagues). Please consider adjusting the framing of the clinical problem.
2. The organization of the introduction with respect to the biomarkers tested is not particularly cohesive, and the paragraphs read more like a bullet point list. Consider reformatting to make the information easier to assimilate to the reader. An informational supplemental table may also be of benefit.
3. Additional discussion of prior studies attempting to improve upon pathologic biomarkers for diagnosing BE and HGD/EAC is warranted to properly frame the present study, including a discussion of knowledge gaps from prior studies.
Methods:
1. Please clarify the statement that the male to female ratio was 3.4:1.96 (line 149)
2. Please provide references for the histologic patterns (cytoplasmic perinuclear, nuclear, other) for staining positivity of the various biomarkers (e.g., MUC2, TFF3, others)
3. Please indicate in the methods whether one or multiple individuals stained and/or interpreted the results of the IHC. If multiple interpreters, ideally there should be a representative validation set of patients where all individuals were blinded and determined to have good inter-rater reliability, which is potentially cited as an issue for missed HGD/EAC in prior papers on this subject.
Results:
1. Please provide some baseline clinicopathologic characteristics of the cohort at large, including at least age, sex, race, and some disease-relevant data if available (duration of reflux symptoms for example). If there are significant differences in baseline characteristics between the cohort at large and any of the sub-groups (BE-IND progressors, non progressors), this should be noted.
2. The bolding for table 3 is confusing without a clear reason for why certain lines are bolded. I suspected this is only for cases with reference to BE-IND, but that is already separately indicated by the two asterisks.
3. It appears that a statistician is among the authorship, however there are a few inaccuracies/oddities that I have noted:
a. I don’t believe that a Fisher exact test is the appropriate test to pair with a relative risk measure of association. To illustrate this, there is at least once instance where the Fisher p value is significant, but the RR 95% confidence intervals crosses 1 (see Table 3, C&N and loss for LGD under beta catenin). Please pair p values with their appropriate statistical test result in Table 3 and throughout the manuscript.
b. In the text, there are similar instances where the p value given is surprising relative to the confidence interval (line 390: 95% CI 1.1-55.6, p=0.0058).
c. Odds ratio is at times used in the text rather than relative risk, please provide a justification for why OR is used, and what test was used to calculate this (logistic regression versus other).
d. The wide confidence intervals for many measures of association (e.g., RR 3-170) suggests that there may insufficient events of interest to run some of these tests.
e. Table 4: there are similar instances where the 95% confidence interval crosses 1 but the p value given is significant.
4. Please show representative FISH images as a figure. Was there any measure of association done of the ability of FISH to predict any outcomes of interest or diagnose any of the disease states of interest? As currently presented, the FISH data do not contribute meaningfully to the manuscript and could be cut to streamline/focus the manuscript without losing meaningful content (as an alternative).
Discussion:
1. Please avoid directly using the brand name of commercial entities/products (TissueCypher) when possible.
2. The authors are encouraged to review the discussion for syntax/grammar errors or unclear diction. For example, line 483: “In addition, FISH testing demands technical challenges to implement as routine laboratory tests.”
3. Line 495 – The authors do not report on inter-observer reliability/variability and are unable to conclude that it is affected by an expanded panel of IHC tests for BE.
4. The flow of the discussion section is a little confusing. In line 495, the authors appear to be starting a conclusion paragraph, but there are two additional paragraphs that follow it. The authors may consider re-organizing the paragraphs within the discussion for improved readability.
5. Please expand considerably on the limitations paragraph.
Comments on the Quality of English LanguageEnglish language quality is good. Some issues with flow/readability that could be improved but this is mostly stylistic.
Author Response
Responses to the reviewer 1 comments:
In the present study, Dr. Choi and colleagues conduct an in-depth study looking at IHC and FISH to diagnose Barrett’s esophagus, provide accurate grading, and prognosticate progression to higher grades. This is an important area of study and the authors are applauded on their efforts in this space. I have several suggestions/comments to improve the manuscript:
Abstract:
- Please reorganize the abstract into a structured format, to be consistent with author guidelines for this journal (Background, Methods, Results, Conclusion).
Response: Revised as requested line 20-39
- Please ensure that all acronyms in the abstract are defined within the abstract itself
Response: Done
Introduction:
- The premise statement that 20% of HGD/EAC diagnoses are missed in ~20% of the antecedent surveillance biopsy within a year is a difficult assertion to make based on available data, and is also different than the clinical question at hand (e.g., what can be done lower the rate of missed BE, and HGD/EAC diagnoses on surveillance endoscopy, which is at most 5% in large population-based analyses such as the one by Dhaliwal and colleagues). Please consider adjusting the framing of the clinical problem.
Response: rephrased /revised : line # 55-67.
- The organization of the introduction with respect to the biomarkers tested is not particularly cohesive, and the paragraphs read more like a bullet point list. Consider reformatting to make the information easier to assimilate to the reader. An informational supplemental table may also be of benefit.
Response: Revised : introduction of biomarker are added in line 97-99. We did not think to add any supplemental tables.
- Additional discussion of prior studies attempting to improve upon pathologic biomarkers for diagnosing BE and HGD/EAC is warranted to properly frame the present study, including a discussion of knowledge gaps from prior studies.
Response: Revised: line # 141-1432
Methods:
- Please clarify the statement that the male to female ratio was 3.4:1.96 (line 149)
Revised: line 162
- Please provide references for the histologic patterns (cytoplasmic perinuclear, nuclear, other) for staining positivity of the various biomarkers (e.g., MUC2, TFF3, others)
Response: The reference(s) on each biomarker describes mostly positive or negative reaction, not in detailed patterns. The detailed description of reaction pattern of each marker is interpreted and described by the author, Y Choi. Dr Y Choi is a board certified immunopathologist and has extensive experience in immunopathology. She was the program director of immunopathology fellowship program and implemented many IHC labs at various institutions.
- Please indicate in the methods whether one or multiple individuals stained and/or interpreted the results of the IHC. If multiple interpreters, ideally there should be a representative validation set of patients where all individuals were blinded and determined to have good inter-rater reliability, which is potentially cited as an issue for missed HGD/EAC in prior papers on this subject.
Response: positive tissue controls and strong positive reaction of HGD cases as representative validation. # Line 200-201
Results:
All -p-value and OR have been eliminated in Tables 3 and 4 and in the texts ( see responses by the Dr Pollack, 3 a-e)).
- Please provide some baseline clinicopathologic characteristics of the cohort at large, including at least age, sex, race, and some disease-relevant data if available (duration of reflux symptoms for example). If there are significant differences in baseline characteristics between the cohort at large and any of the sub-groups (BE-IND progressors, non progressors), this should be noted.
Response: There were significant differences in baseline characteristics between the cohort at large. Inserted in line 185
- The bolding for table 3 is confusing without a clear reason for why certain lines are bolded. I suspected this is only for cases with reference to BE-IND, but that is already separately indicated by the two asterisks.
Response: bold in Table 3 has been removed.
- It appears that a statistician is among the authorship, however there are a few inaccuracies/oddities that I have noted:
- I don’t believe that a Fisher exact test is the appropriate test to pair with a relative risk measure of association. To illustrate this, there is at least once instance where the Fisher p value is significant, but the RR 95% confidence intervals crosses 1 (see Table 3, C&N and loss for LGD under beta catenin). Please pair p values with their appropriate statistical test result in Table 3 and throughout the manuscript.
Simcha Response:
The reviewer makes a good point regarding the appropriateness of the Fisher test in this context. As the modern approach is to emphasize confidence intervals and not use p-values we will eliminate p-values and rely on “significance” by whether the confidence interval contains “1.”. Thus p-value in table 3 and 4 and texts in the results have been eliminated. OR has been eliminated and replaced with RR
Statistical methods have been revised # line 252-256
- In the text, there are similar instances where the p value given is surprising relative to the confidence interval (line 390: 95% CI 1.1-55.6, p=0.0058).
Simcha response:
The changes in “a” above would resolve this issue.
- Odds ratio is at times used in the text rather than relative risk, please provide a justification for why OR is used, and what test was used to calculate this (logistic regression versus other).
Simcha response:
We agree to use the RR consistently. The statistics of Table 4 have been recalculated using the RR
- The wide confidence intervals for many measures of association (e.g., RR 3-170) suggests that there may insufficiently events of interest to run some of these tests.
Simcha response:
With the use of RR, the CI have gotten narrower in Table 4
- Table 4: there are similar instances where the 95% confidence interval crosses 1 but the p value given is significant.
Simcha response:
The changes in “a” above would resolve this issue
- Please show representative FISH images as a figure. Was there any measure of association done of the ability of FISH to predict any outcomes of interest or diagnose any of the disease states of interest? As currently presented, the FISH data do not contribute meaningfully to the manuscript and could be cut to streamline/focus the manuscript without losing meaningful content (as an alternative).
Response: Thank you for your comments. We tested the FISH on 48 patients in a period of several years while the FISH testing has been highly advertised to the gastroenterologists.
Although the FISH testing may be outdated and our findings do not support any value in BE, we like to include the FISH test results as some of the gastroenterologists still order the tests and commercial laboratories still provide the costly tests.
Discussion:
- Please avoid directly using the brand name of commercial entities/products (TissueCypher) when possible.
Response: deleted # line 505
- The authors are encouraged to review the discussion for syntax/grammar errors or unclear diction. For example, line 483: “In addition, FISH testing demands technical challenges to implement as routine laboratory tests.”
Response: revised line # 502-503
- Line 495 – The authors do not report on inter-observer reliability/variability and are unable to conclude that it is affected by an expanded panel of IHC tests for BE.
Response: line 520 has been rephrased
- The flow of the discussion section is a little confusing. In line 495, the authors appear to be starting a conclusion paragraph, but there are two additional paragraphs that follow it. The authors may consider re-organizing the paragraphs within the discussion for improved readability.
Response: reorganized # line 514524
Please expand considerably on the limitations paragraph.
Response: revised # line 526-529
Comments on the Quality of English Language
English language quality is good. Some issues with flow/readability that could be improved but this is mostly stylistic.
Submission Date
01 May 2024
Date of this review
02 May 2024 22:03:34

Reviewer 2 Report
Comments and Suggestions for Authors
Dear Authors,
Thanks for your effort in this field.
It is an interesting study.
BE is now considered a precancerous lesion of EAC. Therefore, an accurate diagnosis and evaluation of BE is necessary. Therefore, this study is of interest. However, there are several issues that need to be pointed out and discussed.
1. A subset of patients in EAC will have overexpression or mutations in Her-2, so why were IHC experiments for Her-2 not performed?
2. On what basis were these antibodies included by the investigators selected?
3. The FISH assay is, in fact, a little bit outdated and will fail to detect more detailed mutations; how do you consider this flaw?
Author Response
Responses to Reviewer 2 comments:
Dear Authors,
Thanks for your effort in this field.
It is an interesting study.
BE is now considered a precancerous lesion of EAC. Therefore, an accurate diagnosis and evaluation of BE is necessary. Therefore, this study is of interest. However, there are several issues that need to be pointed out and discussed.
- A subset of patients in EAC will have overexpression or mutations in Her-2, so why were IHC experiments for Her-2 not performed?
Response: The aim of our study was to identify routinely applicable diagnostic biomarkers of BE and that of progressor. We thought that Her 2 testing is mostly the targeted therapeutic biomarker and was not tested.
- On what basis were these antibodies included by the investigators selected?
Response: As cited in the references, only single or combination of a few biomarkers have been tested by previous authors for the neoplastic progression.
However, we noticed two issues:
- The diagnoses of BE have been missed in BE and BE -associated dysplasia
- The molecular processes of GERD-BE-EAC processes are complex process, single biomarker testing may not provide adequate information.
Thus, we were interested in testing a panel of biomarkers to identify relatively specific and sensitive routinely applicable biomarkers for the diagnosis of BE and neoplastic progression of BE.
- The FISH assay is, in fact, a little bit outdated and will fail to detect more detailed mutations; how do you consider this flaw?
Yes, FISH testing is not widely used as in the past but still gastroenterologists order the tests via commercial laboratories. The FISH testing in our study does not provide any additional information and is technically challenging and should not be used for all BE cases routinely.
In addition, adequate sampling and accurate testing and interpretation of the FISH test could be an issue to adopt as a routine clinical laboratory test.

Round 2
Reviewer 1 Report
Comments and Suggestions for Authors
If one looks at the revised manuscript, there are fewer marked changes within it than the length of the text of my initial reviewer comments. That is almost never a good thing, as a rule of thumb. Additionally, there were several items that I requested for the authors to “show their work” that were not done. Perhaps I did not clearly communicate how extensive I felt the changes to the manuscript needed to be prior to accepting this paper, so I will endeavor to be more clear here, using my prior comments as a guide.
Abstract:
- It was reorganized as requested, but there remains a p value without a measure of association. I feel that both should be present, and that the measure of association should be appropriate to the p value (confidence intervals should not cross 1, etc.)
Introduction:
- Regarding the premise statement that “20% of HGD/EAC diagnoses are missed in ~20% of the antecedent surveillance biopsy within a year.” My comment was not that the statement was unclear, but that this is a false statement of significance. There is no way to logically tie this statement to the central thesis of the paper that improved pathologic testing can decrease this rate. Even if pathology was 99% accurate for esophageal biopsies, 20% of new ECA diagnoses could still have missed HGD because you’re starting from the endpoint (false-negatives). The authors should either rework this introduction as requested, or acknowledge that no data exist to speak directly to this.
- The biomarker section of the introduction is largely unchanged apart from the two lines the authors added. It is choppy, does not thematically progress from one paragraph to the next, and is not suitable for publication in its current form. Please rewrite it.
- Regarding additional discussion of prior studies (last paragraph), the changes the authors have made are very inadequate. There must be multiple citations of prior studies and a brief summary of the efforts of prior efforts to improve pathologic diagnosis on outcomes. A comprehensive introduction of prior papers that have tried to do similar things is an essential component of any introduction and it appears to be absent here.
Methods:
4. With respect to my request that the authors provide references for the histologic patterns for staining positivity, the authors have essentially responded that the author is an immunopathologist and that therefore no citation is needed. That is incorrect, and the authors’ unwillingness to show their work or cite a standard for the measurement/reporting of their data is concerning. Unfortunately, I must now request that the authors include supplemental figures showing an example of a positive result from each of their IHC stains. Specifically, p53, MCM2, p16, Beta catenin, Cyclin D1, and any others for which data was collected.
- Please indicate in the methods whether one or multiple individuals stained and/or interpreted the results of the IHC. This was not done as far as I can tell.
Results:
- Please provide some baseline clinicopathologic characteristics of the cohort at large, including at least age, sex, race, and some disease-relevant data if available (duration of reflux symptoms for example). This was not done as I previously requested. This is standard practice.
- The authors chose to remove all p values from the paper, which was not what I asked. I asked them to reconcile why there were p values that were reportedly significant while the 95% confidence intervals did not cross 1, which is not possible. This is concerning from a data presentation/analysis standpoint. I would like all measures of association and confidence intervals to have associated p values so that I can evaluate the statistical integrity of this submission.
- Please show representative FISH images as a figure. This was not done as I previously requested.
Author Response
Responses to the comments by Reviewer 1 2nd round:
we made major changes in the text and references ( with track changes noted) and added supplemental figures (1&2)to respond to the Reviewer 1, 2nd round comments.
Abstract:
- It was reorganized as requested, but there remains a p value without a measure of association. I feel that both should be present, and that the measure of association should be appropriate to the p value (confidence intervals should not cross 1, etc.)
Simcha response: P- value in the abstract is removed
Introduction:
- Regarding the premise statement that “20% of HGD/EAC diagnoses are missed in ~20% of the antecedent surveillance biopsy within a year.” My comment was not that the statement was unclear, but that this is a false statement of significance. There is no way to logically tie this statement to the central thesis of the paper that improved pathologic testing can decrease this rate. Even if pathology was 99% accurate for esophageal biopsies, 20% of new ECA diagnoses could still have missed HGD because you’re starting from the endpoint (false-negatives). The authors should either rework this introduction as requested, or acknowledge that no data exist to speak directly to this.
Response : The key point in this section of the introduction is the importance of identifying BE to avoid missed opportunity for surveillance and a potential missed opportunity in detecting EAC. Thus, I rewrote the entire section by adding new references and revising references. The missed HGD may have tied to the missed diagnosis of BE due to missed surveillance ( # line 52 to 100).
- The biomarker section of the introduction is largely unchanged apart from the two lines the authors added. It is choppy, does not thematically progress from one paragraph to the next, and is not suitable for publication in its current form. Please rewrite it.
Response: I rephrased and revised ( # lines 131- 196).
- Regarding additional discussion of prior studies (last paragraph), the changes the authors have made are very inadequate. There must be multiple citations of prior studies and a brief summary of the efforts of prior efforts to improve pathologic diagnosis on outcomes. A comprehensive introduction of prior papers that have tried to do similar things is an essential component of any introduction and it appears to be absent here.
Response: I revised and added references on previous studies ( # lines 198-215).
Methods:
- With respect to my request that the authors provide references for the histologic patterns for staining positivity, the authors have essentially responded that the author is an immunopathologist and that therefore no citation is needed. That is incorrect, and the authors’ unwillingness to show their work or cite a standard for the measurement/reporting of their data is concerning. Unfortunately, I must now request that the authors include supplemental figures showing an example of a positive result from each of their IHC stains. Specifically, p53, MCM2, p16, Beta catenin, Cyclin D1, and any others for which data was collected.
Responses: I inserted all pertinent references describing the interpretation of IHC reaction pattern of each marker in the “IHC study section 2.2. (, line 281- 282). I also prepared supplemental figure 1 of HGD cases. However, these were already embedded in figures I-3 in the main MS. I do not think the supplemental figure is needed and seem redundant.
- Please indicate in the methods whether one or multiple individuals stained and/or interpreted the results of the IHC. This was not done as far as I can tell.
Reponses: I edited this in the material and methods: IHC staining in line 276-278 and interpretation of IHC in the lines of 289-292.
Results:
- Please provide some baseline clinicopathologic characteristics of the cohort at large, including at least age, sex, race, and some disease-relevant data if available (duration of reflux symptoms for example). This was not done as I previously requested. This is standard practice.
Bedford response: Line 233-234. We did not analyze the information on race or ethnic background, and smoking history.
- The authors chose to remove all p values from the paper, which was not what I asked. I asked them to reconcile why there were p values that were reportedly significant while the 95% confidence intervals did not cross 1, which is not possible. This is concerning from a data presentation/analysis standpoint. I would like all measures of association and confidence intervals to have associated p values so that I can evaluate the statistical integrity of this submission.
Simcha response: The reviewer correctly points out the inappropriateness of the Fisher Exact test as a measure of association in this context. The problem we are facing is that a p-value is not generated by SAS when it calculates the RR (email from SAS support: “As mentioned, proc freq does not compute the P-Values for relative risk.”) Modern statistical practice (see for example, More Confidence Intervals and Fewer p Values (jacc.org) or P value ban: small step for a journal, giant leap for science | Science News) accepts that the 95% confidence interval is an improvement over the p-value in that in addition to the binary significant/not significant (by the in/exclusion of ‘1’), it crucially gives a measure of the uncertainty of the parameter (RR) estimate. It captures the uncertainty inherent in sampling and quantifies the random error associated with our estimate. Indeed, the p-value is sometimes misleading in that it is partly a function of the sample size and thus a small effect can look like a “highly significant” result.
In the revised manuscript, we Appended “+” to those CI which do not contain 1, as a visual aid to the reader to quickly ascertain significance at the 5% level( Tables 3 & 4) .
- Please show representative FISH images as a figure. This was not done as I previously requested.
Response: I prepared supplemental figure 2 with legend on one patient who BE-IND and had 3 F/U FISH testing.

Round 3
Reviewer 1 Report
Comments and Suggestions for Authors
I thank the authors for their significant edits in this last round. The manuscript has improved considerably and I feel it is now suitable for publication.